# Application of Cone Beam Computed Tomography in Risk Assessment of Lower Third Molar Surgery

**DOI:** 10.3390/diagnostics13050919

**Published:** 2023-03-01

**Authors:** Yiu Yan Leung, Kuo Feng Hung, Dion Tik Shun Li, Andy Wai Kan Yeung

**Affiliations:** 1Oral and Maxillofacial Surgery, Faculty of Dentistry, The University of Hong Kong, Hong Kong SAR, China; 2Oral and Maxillofacial Radiology, Applied Oral Sciences & Community Dental Care, Faculty of Dentistry, The University of Hong Kong, Hong Kong SAR, China

**Keywords:** cone-beam computed tomography (CBCT), inferior alveolar nerve injury, risk assessment, root resorption, third molar surgery

## Abstract

Risks of lower third molar surgery like the inferior alveolar nerve injury may result in permanent consequences. Risk assessment is important prior to the surgery and forms part of the informed consent process. Traditionally, plain radiographs like orthopantomogram have been used routinely for this purpose. Cone beam computed tomography (CBCT) has offered more information from the 3D images in the lower third molar surgery assessment. The proximity of the tooth root to the inferior alveolar canal, which harbours the inferior alveolar nerve, can be clearly identified on CBCT. It also allows the assessment of potential root resorption of the adjacent second molar as well as the bone loss at its distal aspect as a consequence of the third molar. This review summarized the application of CBCT in the risk assessment of lower third molar surgery and discussed how it could aid in the decision-making of high-risk cases to improve safety and treatment outcomes.

## 1. Introduction

Third molar surgery is known to be the most common oral surgical procedure [1]. Lower third molar impaction is common and often leads to oral diseases like pericoronitis, dental caries, localised periodontal disease or pathologies like benign cysts or tumours [2]. Surgical removal of the causative third molar is the only treatment for these conditions, and prophylactic removal is justified if the pathologies are expected to develop over time [3]. Apart from normal surgical risks like post-operative pain and swelling, there are several potential risks that are specific to lower third molar surgery. Neurosensory deficit because of trigeminal nerve injury is a potential long-term complication of lower third molar surgery [4,5,6,7]. The inferior alveolar nerve (IAN), a branch of the mandibular branch of the trigeminal nerve, is particularly at risk because of its proximity to the lower third molar root(s). Anatomically, IAN enters the mandibular foramen at the medial aspect of the mandibular ramus, travels in the inferior alveolar canal (IAC) within the mandible, and exits at the mental foramen to give an ipsilateral cutaneous sensation of the lower lip. Depending on the depth of the impacted lower third molar as well as the course of the IAC, some IAN can touch the root of the third molar or even grooves onto the root at its course. The force generated during the root elevation process by the instrument may inevitably cause an indirect compression onto the IAN and results in a crush injury of the nerve. The rotary instrument for bone removal or sectioning of the root may also traumatise the IAN. Different degrees of IAN injury present with different symptoms and duration of neurosensory deficit, of which a proportion could be permanent. It is also proved that on top of the sensory changes and potential chronic neuropathic pain, trigeminal nerve injury from lower third molar surgery could lead to reduced quality of life and even depression in the affected individuals [8,9,10]. Prediction of nerve injury risk and consideration of preventive measures are, therefore, of the highest importance for the pre-operative third molar surgery assessment. In many countries, there is also a medico-legal implication to provide sufficient risk assessment in order to give thorough informed consent for the surgical procedure [11,12]. 

One major indication of lower third molar surgery is to preserve the adjacent second molar, which could be affected by the third molar impaction because of persistent food trapping and recurrent infection [13]. There is a need to assess the risk of existing periodontal bone loss as well as to predict subsequent periodontal attachment regeneration. Persistent eruption of an impacted third molar onto the root surface of the adjacent second molar may cause root resorption and potentially create a portal of microorganisms that leads to periapical infection of the functional second molar. With all these considerations in mind, appropriate imaging modality is essential in the risk assessment of lower third molar surgery. Conventional imaging by 2D plain radiographs has served the purpose for many years but with limitations. With the development of 3D imaging like the dental cone beam computed tomography (CBCT), additional information can now be acquired. Considerations of the risks and benefits of the advanced imaging technology should be taken, in particular, the concern of additional radiation dosage of CBCT. The “As Low As Reasonably Achievable” (ALARA) principle in 1977, set by the recommendations of the International Commission on Radiological Protection, still holds as the gold standard in this aspect [14]. Supported by recent literature, this narrative review aimed to discuss the use of CBCT in the risk assessment of lower third molar surgery and to compare its advantages over the traditional imaging method. 

## 2. Conventional Risk Assessment of Inferior Alveolar Nerve Injury 

Orthopantomogram (OPG) has been the most useful imaging for the assessment of IAN injury risk of lower third molar surgery [15,16]. OPG is a panoramic scanning dental radiograph that shows a flattened 2D view from one side of the jaw to the other side. Apart from showing the mandible and the maxilla, it also shows the maxillary sinuses, the nasal cavity and the orbits, and for this reason, its applications cover almost all scopes of dentistry [17,18,19,20,21]. OPG is relatively cheap and easily available, with good imaging quality, in particular, at the region around the posterior mandible [22]. The risk of IAN injury is directly related to the proximity of the IAC and the lower third molar root. IAC is a corticated bony canal that harbours the IAN [23]. The understanding of this specific anatomical structure is twofold: (1) The two white lines on the OPG that represent the IAC are formed by the upper and lower cortices of the bony canal since the x-ray is absorbed by the denser bone; (2) The proximity of the third molar root to the IAC will show specific radiographic signs because the canal cortex is bleached or deviated from its course [24,25]. It has been a topic of interest to identify specific radiographic signs of OPG. “Darkening of the root” as a radiographic sign on OPG was consistently found to be related to IAN injury and IAN exposure after third molar removal [24,25,26,27,28] (Figure 1). Leung and Cheung evaluated 178 OPGs with lower third molars showing one or more radiographic signs that indicated close proximity to the IAC and found “darkening of root” and “displacement of the IAC by the root were significantly related to IAN exposure, but only “darkening of root” showed a significant risk of IAN injury. The group also found two or more radiographic signs also showed an increased risk of IAN injury [24]. Other radiographic signs also showed an increased risk of IAN injury but with less consistency [26,29]. Su et al. concluded in their systematic review and meta-analysis that specific radiographic signs of OPG could be considered sufficient for ruling out the risk of postoperative IAN injury but not for ruling out postoperative IAN injury [29]. A meta-analysis by Liu et al. also concurred with the view and concluded despite the interpretation of OPG on “darkening of root” had high specificity in predicting IAN injury after lower third molar surgery, its ability to detect true positive IAN injury was not satisfactory [30]. 

## 3. CBCT vs. OPG in Assessment of IAC Proximity

Dental CBCT machines were developed and commercialised in the late 1990s. Since then, CBCT has been popularized, and it is used extensively in the field of dentistry. Its utilization was largely advanced by the development of dental implantology [31]. With CBCT, the width and the height, as well as the bone quality of the alveolar bone available for dental implants, can be clearly visualised. Important anatomical structures like the IAC and the maxillary sinuses are important landmarks that should also be visualised and avoided in dental implant surgery, which can be clearly demonstrated in CBCT [32]. 

In third molar surgery, CBCT offers a thorough understanding of the 3D relation between the third molar and the IAC [33]. The bone and tooth structure in cross-sectional images can be clearly displayed in CBCT, as well as their spatial relationship. Moreover, the bucco-lingual position of the overlapping hard tissue structures can be demonstrated, as well as the type of third molar impaction and the specific number and morphology of the root(s). Since IAN injury is directly related to the contact of the root and the IAC, visual confirmation by a coronal CBCT image to demonstrate the true contact between the two structures is the solid proof of their relationship (Figure 2). Tantanapornkul et al. noted that CBCT carried a sensitivity and specificity of 93% and 77%, respectively, to predict IAN exposure, which was significantly higher than OPG [34]. Hasani et al. performed a prospective study to compare CBCT and OPG in predicting IAN exposure after third molar surgery and found that CBCT was accurate in 93.3% of the cases, which was much higher than the 67.7% by OPG [35]. Reia et al. performed a systematic review and meta-analysis and confirmed that CBCT was superior in predicting IAN exposure [36]. 

CBCT undoubtedly offers a better visual of the relationship between the tooth root and the IAC as an imaging modality. Yet, does the difference lead to clinical relevance? The query was raised by several researchers who argued the absolute need for routine CBCT as a pre-operative assessment imaging tool. Matzen et al. prospectively evaluated 186 lower third molar surgery and found that only 12% of the cases had changed the treatment plan as an influence by the CBCT [37]. Manor et al. concurred with the view and summarised that treatment decisions of lower third molar surgery could be accepted without CBCT findings, and there was little effect of the information from CBCT on the treatment decision of the surgical intervention in comparison to OPG [38]. A systematic review concluded that despite CBCT being better than OPG in visualising the tooth root and IAC proximity, there was no influence on the choice of surgical technique on the removal of the third molar [39]. The key factor of whether using CBCT could prevent IAN injury is debatable. Korkmaz et al. found that CBCT could reduce the prevalence of temporary IAN deficit after lower third molar surgery when compared to OPG, but no significant difference in permanent IAN deficit [40]. Guerrero et al. performed a randomized clinical trial and concluded that although CBCT could be more accurately confirming the bucco-lingual position of the IAC when compared to OPG, it was not better in preventing IAN injury [41]. A recent systematic review and meta-analysis by de Toledo Telles-Araújo et al. found that there was no statistically significant difference between the CBCT group and OPG group in the prevalence of neurosensory deficit after lower third molar surgery and therefore concluded that CBCT was not superior to OPG in avoiding IAN injury in lower third molar surgery [42].

Compared to OPG, CBCT carries a larger radiation dose. For this reason, the European Academy of Dento-Maxillo-Facial Radiology stated that CBCT should not be used in a routine manner for the assessment of lower third molar surgery and should be prescribed when plain radiography could not provide sufficient diagnostic information [43]. Cost and availability are also considerations when determining whether CBCT should be taken for the assessment of lower third molars. Yeung et al. showed that only 50% of the dental clinics were installed with a CBCT machine in Hong Kong, an international city that may represent other metropolitan cities of a developed country [44]. The need to order a CBCT should be clearly justified, for instance, when OPG demonstrates the third molar is near the IAC. One of the recent breakthroughs in the management of lower third molars with high IAN injury risk is proving that the coronectomy is a safe alternative to total removal of the lower third molar [45,46,47,48,49,50]. Coronectomy involves removing the crown of the tooth and intentionally retaining the root, which protects the IAN and removes the infection cause [45]. The retained root will migrate but usually stay within the alveolar bone [51,52]. There is also some evidence to show that the bone regeneration at the distal aspect of the adjacent second molar appears to be better than the total removal of the lower third molar [53,54]. Additional information from CBCT allows the clinicians to make informed consent and clinical decision with the patient on the third molar surgery, in particular on the expected risk of IAN injury and the option of coronectomy instead of total removal [7]. With a lower third molar with moderate-to-high risk of IAN injury, as shown on CBCT, it is reasonable to choose coronectomy over conventional total removal for safety concerns.

## 4. Dosage of CBCT: What’s Optimal

One of the major drawbacks of CBCT is its radiation exposure and the related potential health hazards [55]. Although a CBCT scan has a lower radiation dose than a medical-grade spiral CT scan, it is still higher than an OPG [56]. The key to IAN risk assessment is the visibility of the IAC. Researchers tried to test the different settings of CBCT in order to reduce the radiation dosage of CBCT but can visualise the IAC. Exposure variables, such as tube current, tube voltage, scanning time and the size of the field of view (FOV), can be altered to adjust the radiation dose while balancing a sufficient image quality [57,58,59]. Protocols for a lower dosage have been suggested for CBCT examinations without the loss of important image information [60]. Ex-vivo studies on dry human mandibles with low-dose or “ultra” low-dose protocols were performed to investigate the ideal settings to visualise the IAC with reasonable qualities [59,61,62]. Zaki et al. showed that low-dose CBCT of 49.6 mGy×cm^2^ could clearly exhibit the IAC at the mesial root of the third molar [61]. However, it is understood that these experiments might not represent the clinical scenario because of the lack of soft tissue and the presence of collapsed trabecular bone. Nonetheless, the trend of reducing radiation dosage to acquire diagnostically acceptable quality is rising. The hope of reducing the radiation dosage of a CBCT similar to or even lower than the recommended dose of 80 to 139 mGy×cm^2^ for OPG could be fulfilled with well-designed clinical studies [63,64]. The improved safety with better imaging information by low-dose CBCT could potentially replace OPG as the routine pre-operative risk assessment for lower third molar surgery in the future.

## 5. CBCT for Assessment of External Root Resorption of Adjacent Second Molar

Apart from pathologies like caries or pericoronitis, the impaction of third molars may result in external root resorption (ERR) of the adjacent second molars. It was found that 20–47.7% of the impacted lower third molars are associated with ERR of adjacent second molar [65,66,67,68]. A recent cross-sectional study by Lacerda-Santos et al. evaluated 107 CBCT scans and found that male patients had more ERR of the adjacent second molars when compared to their female counterparts, and the probability that ERR would affect the second molar was 1.71 times greater when the third molar exhibited the Pell and Gregory class IC position and 1.64 times greater when the third molar exhibited the Winter mesioangular position [68]. Suter et al. reported similar findings of male predilection and the mesioangular impaction of the lower third molar as risk factors, with the majority of ERR of the adjacent second molar considered to be slightly resorbed [66]. The studies on ERR of adjacent second molars all concluded that the phenomenon is common and should inform the patients if it is suspected on lower third molar assessment. The aetiology of ERR is believed to be caused by the pressure exerted by the third molar or its dental sac, which stimulates osteoclastic activities onto the root surface of the second molar [69,70,71]. The condition of ERR of a functional tooth by an impacted tooth is not limited to the lower third molar region. A similar finding was noted in the maxillary third molar that causes ERR of the adjacent second molar [72,73]. A large Chinese study by Shi et al. evaluated over 75,000 CBCTs and found that impacted maxillary canines also caused over 25% of root resorption of neighbouring tooth/teeth [74]. ERR of a functional tooth may result in a periodontal defect or an endodontic lesion, which may jeopardize the long-term health of the tooth.

The clinical decision to prophylactically remove the impacted lower third molar to prevent ERR relies on the position of the third molar to the adjacent second molar. Traditionally, OPG is used to assess if there is a risk of ERR of the second molar when the third molar is impacted. However, it was found that the accuracy of OPG in diagnosing ERR was low because of the overlapping 2D images. The use of CBCT to assess ERR of the second molar as a consequence of impacted lower third molar is found to be more accurate. Oenning et al. found that for the same sample of patients with impacted third molars, OPG only found about 23% of the ERR of adjacent second molars that were diagnosed by CBCT [75]. With an accurate assessment of ERR, it is, therefore, possible to correctly identify the risk factors that may contribute to ERR of the second molar. It was found that about half of the horizontally, mesioangularly, or transversely impacted lower third molars are related to ERR of the second molars [66,76]. Wang et al. found that age over 35 years old is an important risk factor for ERR, which is strong evidence to support early prophylactic removal of the lower third molar when the risk of ERR is high [67].

With the presence of an ERR of the adjacent second molar, patients should be notified of the risk of the pulpal pathology of the second molar after the removal of the impacted third molar. Fortunately, the majority of adjacent second molars with ERR by the lower third molar appeared to be unaffected after the lower third molar removal. A classic paper by Nitzan et al. found that in most cases, the damaged periodontium of the second molar as a consequence of ERR was completely re-established one year after the third molar surgery [70]. However, patients should be informed of the potential complications when there is an advanced ERR of the adjacent second molar. Qu et al. reported 9.5% of second molars with ERR presented with signs or symptoms and required treatment or extraction [76,77]. It appeared that older age increased the chance of developing pulpal pathology of the second molars with ERR. The majority of ERR of the second molar did not need further intervention except observation [77].

## 6. CBCT for Assessment of Bone Loss of Adjacent Second Molar

It is known that third molar impaction affects the periodontal health of the adjacent second molar and leads to periodontal attachment loss of the tooth [78]. It is caused by persistent infection because of food trapping and plaque retention, which leads to an inflammatory response and consequential bone and attachment loss. It is important to inform patients with an impacted lower third molar about the prognosis of the adjacent second molar, in particular when there is advanced periodontal bone loss at the distal aspect of the tooth. Removal of the impacted third molar is a treatment of the condition, but the periodontal pocket may persist, especially when there is severe localised bone loss pre-operatively. The dilemma of considering whether to remove the lower third molar or not is often the reason for the procrastination of the treatment. One study found an average of 5.7 mm pocket at the distal aspect of the second molar at nine years post-third molar surgery if there was no adjunctive periodontal treatment [79]. Older age was found to be a risk factor for the residual periodontal pocket. Kugelberg noted that over 25% of patients older than 25 years had a residual pocket of over 7 mm at the distal aspect of the second molar 2 years after the surgery [80]. A recent systematic review and meta-analysis showed baseline periodontal probing depth was strongly correlated with final periodontal probing depth [81]. When the pre-operative periodontal bone loss is too advanced, the long-term prognosis of the second molar could be jeopardized even when the causative third molar is removed. As part of the risk assessment and informed consent procedure of the third molar surgery, the clinician should discuss the fate of the adjacent second molar, considering the amount of periodontal bone loss prior to the surgery. While OPG is usually sufficient as an assessment imaging for many third molar surgery cases, CBCT allows accurate 3D evaluation of the pre-operative and post-operative bone level of the second molar (Figure 3). Dias et al. found that for the same sample of cases with impacted lower third molars, CBCT could detect 80% of the cases with marginal bone loss, while OPG could only detect 62.9% [82]. For post-operative periodontal healing of the second molar, a recent CBCT study on lower third molar coronectomy showed an increase in the bone level at the distal aspect of the second molar ranging from 3.2 mm to 3.5 mm in the long-term follow-up [53]. When extensive bone loss is noticed, guided bone regeneration on the second molar may be performed together with the third molar surgery to improve the periodontal attachment of the tooth [54,81]. With the information from the CBCT, the clinician could inform the patient on the periodontal condition of the second molar, the estimation of its long-term prognosis, as well as treatment alternatives or adjunctive treatment required before and after the lower third molar surgery.

## 7. CBCT Risk Assessment of Lingual Plate Fracture

Lingual plate fracture during lower third molar surgery is uncommon but potentially may lead to other severe complications like lingual nerve injury, bleeding, or displacement of fractured root or tooth into the sublingual space [7,83]. The roots of some lower third molars may lean onto the lingual plate, or some lingual plates could be very thin and prone to fracture when force is applied [84]. Two-dimensional radiographs like OPG could not visualize the thickness of the lingual plate or the buccal-lingual angulation of the third molar impaction. CBCT allows a clear demonstration of the lingual plate thickness and its spatial relation with the third molar root. In some cases, the root may perforate the lingual plate, and with the help of CBCT, the surgeon may be more cautious and exert less apical force during the root elevation. Wang et al. evaluated 364 lower third molars in the Chinese population and found that the majority (72.8%) of the lower third molar roots were in contact with or perforated the lingual plate [85]. Additional safety measures, like raising a larger flap, and additional tooth sectioning with the use of piezosurgery, can be applied to avoid fracturing the lingual plate and pushing the root into the sublingual space [84,85].

## 8. Limitations and Drawbacks of CBCT

Despite the potential benefits of CBCT in assessing the risks of lower third molar surgery, the imaging modality is not without its limitations. Apart from the higher radiation dosage, it is known that CBCT is more expensive in the equipment and maintenance costs, which may not be affordable for some patients. Routine screening of third molars using CBCT is costly and may be considered unnecessary. It is of note that the interpretation of the images of CBCT needs the proficiency and experience of the clinicians, which forms the most critical part of the risk assessment process. Wrong interpretation may put the patients at risk of potentially irreversible damages like IAN injury.

## 9. Conclusions

Radiographic imaging has been the most useful tool for risk assessment of lower third molar surgery. CBCT is useful in understanding the 3D relationship between the third molar and the relevant adjacent structures, in particular, the IAC and the adjacent second molar. CBCT has been proven to be sensitive to identifying the true proximity of the tooth root and the IAC, yet it might not reduce the risk of IAN injury if the third molar is to be removed in a conventional manner. Since coronectomy of the lower third molar is proven to be safe in the long term, it offers a treatment alternative if a third molar carries a high IAN injury risk. CBCT may be useful in the decision-making of whether coronectomy or total removal of the lower third molar shall be performed by considering the proximity of the tooth and the IAC. There are also efforts to reduce the radiation dosage of CBCT to acquire a diagnostically acceptable image to improve the safety of the imaging modality. CBCT is also used to assess the risk of second molar root resorption or bone loss as a consequence of the third molar impaction. CBCT may also identify risky cases of lingual plate fracture when the lingual plate is very thin, or the tooth root perforates the lingual plate. The development and research of CBCT have increased its popularity as part of the risk assessment and informed consent procedure for lower third molar surgery. Further research on low-dose CBCT to acquire similar diagnostic-value images is ongoing to improve patients’ safety.

## Figures and Tables

**Figure 1 diagnostics-13-00919-f001:**
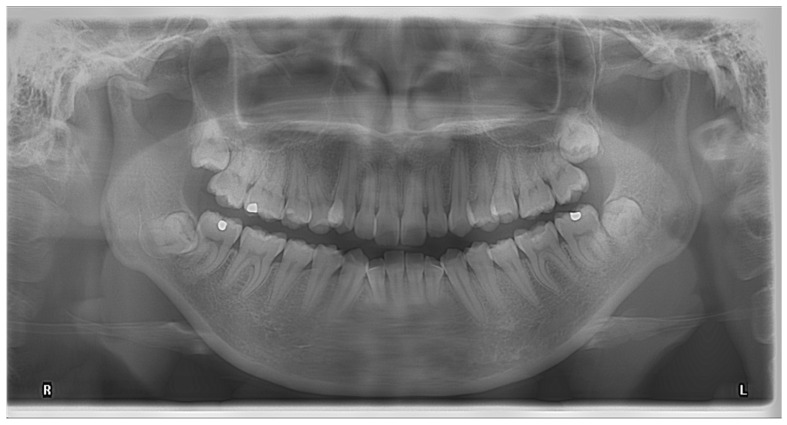
OPG showing the darkening of the root of a right lower third molar.

**Figure 2 diagnostics-13-00919-f002:**
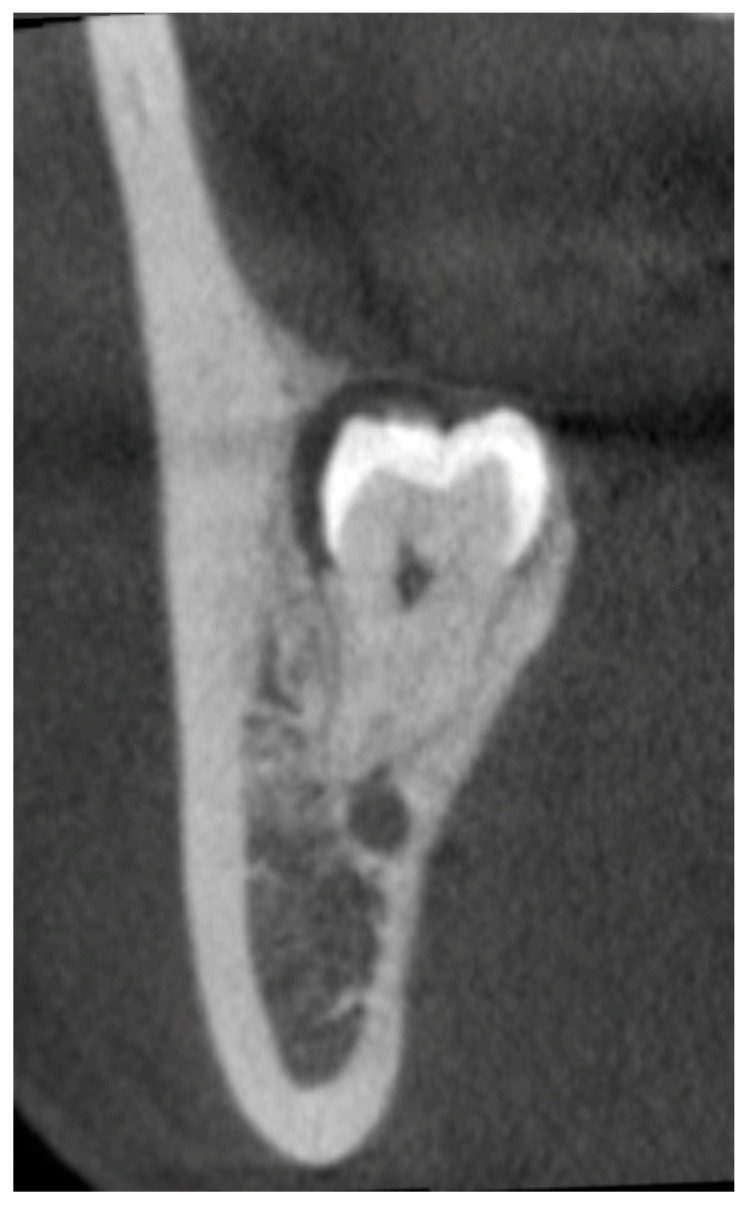
Representative CBCT image showing the loss of the upper cortex of the inferior alveolar canal.

**Figure 3 diagnostics-13-00919-f003:**
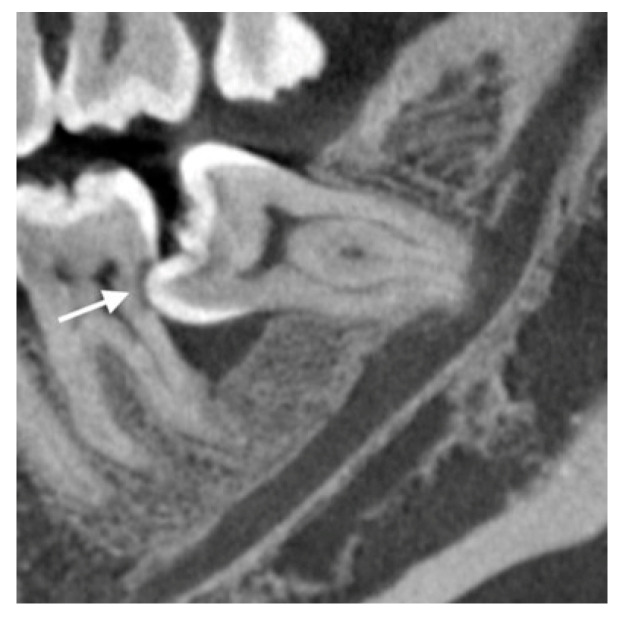
Representative CBCT image showing bone loss at the distal of a left lower second molar with root resorption (white arrow) caused by a mesioangular impacted third molar.

## Data Availability

Not applicable.

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
