# Peer review of "Application of Cone Beam Computed Tomography in Risk Assessment of Lower Third Molar Surgery"

_diagnostics, 2023, doi:10.3390/diagnostics13050919_

Round 1

Reviewer 1 Report

A good review. Minor errors in editing and referencing.

Author Response

Many thanks for the comments. We have made the amendments accordingly.

Reviewer 2 Report

I congratulate the authors for conducting the present study on the use of CBCT as a diagnostic tool for third molars surgery. Here goes a few concerns:

I suggest the authors to place the type of the study design in the title.

I recommend the authors to place the keywords by alphabetic order.

I suggest the authors to debate the ALARA issue on the Introduction. And the recommendations, by specific associations, regarding the recommendations for the use of CBCT, which this is for sure one of them.

On the comparison between the CBCT and OPG is it possible to add an image of the same case seen by both examination (if available)? It would complement very well the advantages of 2D vs 3D.

Figure 1 is good. However is it possible to complement it with a few images to mesial and distal of this case so we can have an idea of the canal pathway in relation to the tooth?

I would like to see a sub-heading, probably just before the conclusions subheading, mentioning a resume of the drawbacks of the CBCT. There are also limitations.

A further research comment would also be welcome.

Please notice that the final references are not according to the journal guidelines.

Author Response

I congratulate the authors for conducting the present study on the use of CBCT as a diagnostic tool for third molars surgery. Here goes a few concerns:

I suggest the authors to place the type of the study design in the title.

Response: We thank the Reviewer’s comment. The authors hope to keep the current title which is in line with the titles of other review papers of the journal.

I recommend the authors to place the keywords by alphabetic order.

Response: Thanks for the suggestion. We have amended accordingly.

I suggest the authors to debate the ALARA issue on the Introduction. And the recommendations, by specific associations, regarding the recommendations for the use of CBCT, which this is for sure one of them.

Response: The authors appreciate the comment. We have added the ALARA debate in the introduction as suggested.

On the comparison between the CBCT and OPG is it possible to add an image of the same case seen by both examination (if available)? It would complement very well the advantages of 2D vs 3D.

Response: Thanks for the suggestion. We apologize that we could not find the correlating OPG image of the same case for this case. Instead, we have inserted a new figure (Figure 1) to show the “darkening of root” sign on an OPG.

Figure 1 is good. However is it possible to complement it with a few images to mesial and distal of this case so we can have an idea of the canal pathway in relation to the tooth?

Response: We apologize that we could not complement it with a few images mesial and sital to this case. We hope the Reviewer is also ok with this limitation we are following.

I would like to see a sub-heading, probably just before the conclusions subheading, mentioning a resume of the drawbacks of the CBCT. There are also limitations.

Response: A sub-section of the limitations and drawbacks of CBCT is now included as suggested.

A further research comment would also be welcome.

Response: We thank the Reviewer for the suggestion. A comment on the future research direction is added in the conclusion.

Please notice that the final references are not according to the journal guidelines.

Response: Thank you for the comment. We have amended according to the journal guidelines.

Round 2

Reviewer 2 Report

Dear authors, I have no more concerns.